# Integrated Value Model to Assess the Sustainability of Active Learning Activities and Strategies in Architecture Lectures for Large Groups

**Oriol Pons [1,*] , Jordi Franquesa [2] and S. M. Amin Hosseini [3]**

[1] Department of Architectural Technology, The Polytechnic University of Catalonia (UPC), Av. Diagonal 649, 08028 Barcelona, Spain

[2] Department of Urbanism and Regional Planning, UPC, Av. Diagonal 649, 08028 Barcelona, Spain; jordi.franquesa@upc.edu

[3] Department of Civil and Environmental Engineering, UPC, Jordi Girona 1-3, 08034 Barcelona, Spain; amin.hosseini@upc.edu

* Correspondence: oriol.pons@upc.edu; Tel.: +34-93-4016391

**Abstract:** At present, university professors lack the tools to know which is the most sustainable activity and/or strategy that should be incorporated into large-group theoretical classes in order to improve our students' learning process whilst taking each scenario into account. These scenarios have different order thinking levels, numbers of students, available time, classroom size and professor skills, among other factors to consider. In architecture schools we have this problem in theoretical lectures. This project has developed and applied a new multi-criteria decision making tool incorporating a mathematical algorithm in order to choose the best set of active learning activities for each case for these lectures in architectural technology courses. This process has relied on seminars involving experts and the use of The Integrated Value Model for Sustainable Assessment. This tool has been very useful to solve the aforementioned problems because architecture professors have been able to choose the most sustainable activity for each scenario considering the alternative sustainability indexes. This first application has been highly useful to assist professors to incorporate active learning methodologies in their classes and to promote lecturers' management of their course contents and time. Future improved versions of this tool will increase its interactivity and broaden its scope.

**Keywords:** MCDM; MIVES; AHP; Knapsack; architecture; active learning; lectures; Bloom Taxonomy

---

## 1. Introduction

Large group lectures are an ancient and traditional way of teaching [1,2] but continue to be an important part of university teaching activities at present. This occurs partly because universities have pragmatic reasons such as student ratios per course and professor, hours per course and space per course, especially in first-year classes [3–5]. An important part of these theoretical classes are given mainly as one-way teaching, in large groups of 100 students or more [6] and have a duration from 50 to 180 min [7]. These historic classes have been analyzed in numerous studies for decades [8,9] from diverse perspectives such as the duration of lectures depending on student attention capacities [10], lectures assessment and its improvement [11], student engagement [12] and classroom space configuration and size [13,14]. The educative community's satisfaction is diverse, with studies in favor [15] and studies against these theoretical classes [16]. One of their main weaknesses are students' passive role and their consequential low learning performance [17] to which some studies suggest new approaches for university courses based on laboratory lessons [18] and group work [19], or online editions [20] could

prove beneficial, while numerous studies and resources suggest the need to introduce active strategies and methodologies [21].

In this sense, the main objectives in this project were: (a) to find the most sustainable active methodologies, tools, activities and strategies to promote deep learning and active roles by students during large group lectures and (b) develop a new tool to help professors choose the most sustainable active learning strategies for large groups attending lectures, taking into account different possible scenarios. This new model has been defined in detail and applied to the "Construction II" course at The Polytechnic University of Catalonia (UPC)-Barcelona Tech. In this sense, this paper presents previous related projects, the specific study case, the research methods for developing the new tool, the results of its application and analyzes these results to achieve first conclusions, recommendations and define future projects.

## 2. Literature Review

There are numerous previous related studies focusing on improving large group lecture learning processes. There are books, articles and numerous conference papers found in technical literature about how to improve large group lectures. These publications collect resources, strategies, methodologies, activities and tricks to improve this type of pedagogical style, dating from the early 1990s. Table 1 classifies a representative sample of the most relevant publications for this research project.

**Table 1.** Classification of scientific articles and books on improving learning processes for large group lectures.

| Main Research Area | References |
|---|---|
| Lecturer experience and training influence on lectures | [22] |
| Guide to organize, manage and teach a large group lecture | [23,24] |
| List of tips and strategies to improve lectures | [4,25–32] |
| Active learning to improve students' participation | [25,33–41] |
| Active learning with limited resources | [42] |
| Advantages of large classes | [9,43] |
| Large classes assessment | [11,44,45] |

There are also numerous universities which have online resources to improve the student learning process in classes which have large student attendance; for instance the University of Bath, University of Waterloo and Vanderbilt University [46–48]. These publications and resources are available to professors so they can study them and follow their recommendations. But this present study has not found in this literature review any index or assessment for these activities depending on numerous aspects that would be interesting to assess simultaneously. Nor have these researchers found any active resources either which can help professors to choose the best strategy or activity for each case considering the specific conditions of each scenario.

## 3. Identification of the Problem

### 3.1. Initial Diagnosis

As said in the introduction, this research paper focuses on architecture schools and departments, where professors commonly lecture nowadays, mainly for theoretical courses. During the history of teaching and learning architecture there have been numerous and different approaches, such as Beaux-Arts and Bauhaus [49]. These approaches are still under debate at present [50] as well as the theory of architecture learning process and its relation to other disciplines [51].

Architecture studies have several areas of knowledge that commonly divide their study plan into courses such as design studio, history, aesthetics and technology, among other topics. This project aims to be applied to a broader sample in the future but has started with the architectural technologies practiced at the Barcelona Architecture School (ETSAB), Polytechnic University of Catalonia (UPC),

because the authors of this project currently are professors and researchers at this institution. In this school, the aforementioned technologies have an important role, not predominant, but balanced in relation to the other areas so that students learn architecture from a holistic point of view [52]. Specifically, this project started with construction courses at this architectural school. In the last century several professors have prepared rigorous materials for these courses [53–56] and have carried out research about the teaching methods used, their problems and possible solutions [57].

This project analyzed the particular case study of "Construction II". This is a compulsory undergraduate third year course, which aims that students understand the importance to incorporate constructive issues which buildings have during their design process. In this sense, this subject includes a high amount of crucial concepts related to architecture construction, from foundations to slab floors, including retaining walls and load bearing systems [58]. This course has four sessions per year, two each semester, one morning shift and one afternoon shift. Each shift has 80 students, with a total of 320 students per year.

This research paper reconsidered the previous learning methodologies this course has employed, which have been three consecutive hours of lectures and two-hour practical sessions. These lectures have had three endemic problems during recent decades: (1) student attendance was really low, (2) students' learning results during these lessons were low and superficial; and (3) students did not participate, did not take an active role. These problems had been detected and confirmed by professors' observations during classes, students' exams and practical exercises and specific research projects, as explained in detail by Pons et al. [59].

The first response in order to solve this problem was the introduction of active learning activities and strategies for theoretical classes designed for large groups, as presented in depth by Pons and Franquesa [60]. The professors who teach this course studied the publications and resources presented in the previous section in order to incorporate active strategies to this subject to solve the previously mentioned three endemic problems. This new strategic approach implied the use of videos, flipped classes, online contests, online questionnaires, theatrical explanations and cooperative activities, among others. The aforementioned incorporation followed an innovative method based on Bloom's Taxonomy revised by Anderson [61]. This implementation collected several indicators about the three endemic problems which obtained satisfying results that still left room for further improvement, as shown in Table 2.

**Table 2.** Some indicators assessed in 2017–2018 academic year and in previous courses.

| Criteria | Indicators | Before 2017–2018 | Course 2017–2018 |
|---|---|---|---|
| Student attendance | Attendance average (%) | 35% | 70% |
| Student satisfaction | Participation (%) | 21% | 34% |
| | Lecturer is a good professor [1] | 2.2 | 3.9 |
| | Lecturer is receptive to students' queries [1] | 2.5 | 4.1 |
| Final results, grades [2] | Students with grade ≥9 | 0% | 3% |
| | Students with grade ≥7 and <8 | 1.8% | 26% |
| | Students with grade ≥5 and <7 | 59.6% | 58% |
| | Students with grade <5 | 24.6% | 3% |
| | Number of students dropping out of the course | 11.2% | 10% |

[1] 0–5 scale, 5 maximum satisfaction; [2] 0–10 scale, 10 maximum satisfaction.

The aforementioned incorporation also collected open satisfaction questionnaires from students regarding the following seven indicators: both students' and professors' dedication both in class and outside, students' satisfaction and attendance and professors' feedback time. The analysis of these questionnaires and seven issues results concluded that the applied active strategies had different strengths and weaknesses, which should be taken into account in future learning methodology applications. For example, this outcome could be considered by lecturers in order to decide the best active strategies for their courses. Another conclusion based on this research was that in order to

integrate each of these seven variables, and even more, into a professor's teaching strategy, a new methodology was necessary, as stated in the introduction to this article.

*3.2. Case Study*

The definition and application of this new tool has focused on the first semester "Construction II" 2018–2019 course, in both its morning and afternoon sessions. This course had 14 instruction sessions, 11 which were 3 h long and three sessions 2 h long. The morning section had 77 students while the afternoon section had 52 students. There were two different professors, one gave nine lectures and the other five.

## 4. Research Methods

This new research project tool to assist lecturers has been defined during two seminars when experts participated, on 22 May 2018 and on 29 October 2018, at which different phases were discussed and the participants agreed to use tools such as analytic hierarchy process (AHP; Appendix A presents a complete list of abbreviations) (Saaty, 1990). These discussions relied on specific technical literature and previous research [60]. These seminars were comprised of members from several institutions: ETSAB university professors with different specialties, the vice director and two students; one La Salle architecture university professor; members from the Gilda research group on innovation in architecture learning and management; members of the UPC Science Education Institute ICE; members of the Education Department Cesire and a pedagogue from the research group Pedagogy, Society and Innovation with the support of the Information and Communication Technologies (PSITIC), part of the Blanquerna Education Faculty. These members were chosen because of their related and complementary expertise and experience in teaching architecture, specifically coordinating and studying at ETSAB, teaching architecture and other disciplines in other universities, carrying out research education and their knowledge about pedagogy respectively. In consequence, these experts were able to assist the authors in defining and improving this new tool, from the general perspective of higher education, research and pedagogy and, at the same time, take into account its first application in architectural studies. These seminars defined the six-phase tool presented in Figure 1, citing the specific actions, actors and methods in each phase.

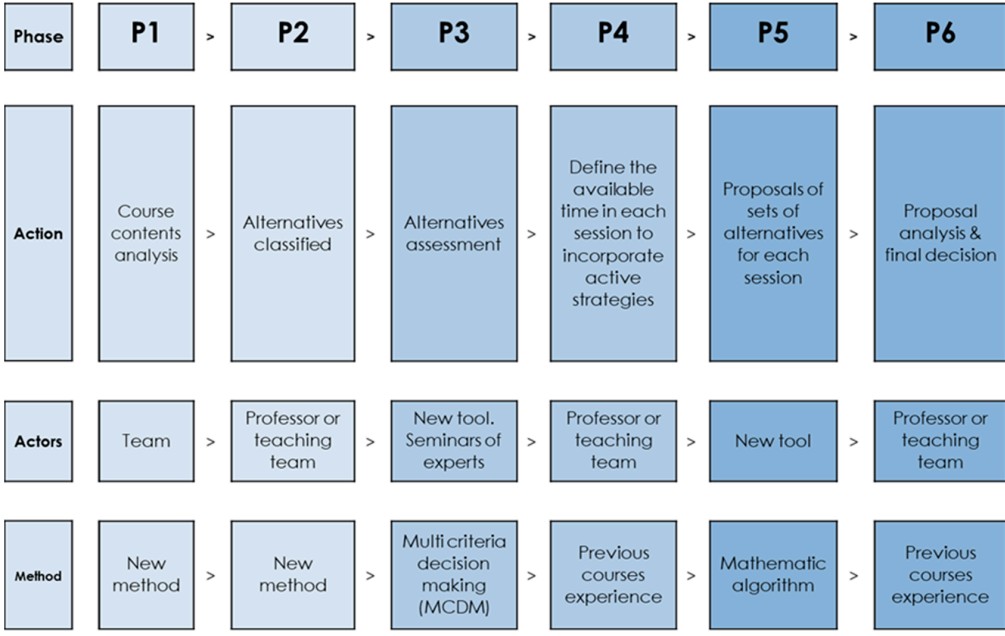

**Figure 1.** Six phase tool to assist lecturers and the actions, actors and methods in each phase.

These six phases are explained in depth in the following paragraphs, while a general synthetic introduction is as follows: during the first two phases, the professor or teaching team analysed the course contents and classified the alternatives following the aforementioned new method; in phase three, the alternatives were assessed in seminars whose participants were experts who use a multi-criteria decision making (MCDM) method in order to be able to take into account multiple indicators; in phase four, the teaching team defined the available time in each session to incorporate active methodologies; in the fifth phase, a mathematical algorithm proposed sets of alternatives so that, in the last phase, the teaching team analysed the given proposals and made its final decision.

## 4.1. Phases 1 and 2: The New Method

The first two phases followed a new method [60], which was introduced in the initial diagnosis section because it was previously defined and applied by the authors in a previous project. These phases assessed the course contents analysis and classified the active alternatives based on Bloom's Taxonomy revised by Anderson. In consequence, this analysis and classification considered three order thinking levels (OTL) included in this taxonomy: lower order thinking level (LOTL), middle order thinking level (MOTL), high order thinking Level (HOTL). These OTLs differ from the thinking complexity required for students which is: low in LOTL—for example remembering and recounting concepts; middle level in MOTL—for instance applying and understanding ideas and information; and high in HOTL—such as analysing, evaluating and creating their own proposals. These first phases are presented in Figure 2, Table A2 summarizes the course contents and Table A3 lists the 25 alternatives.

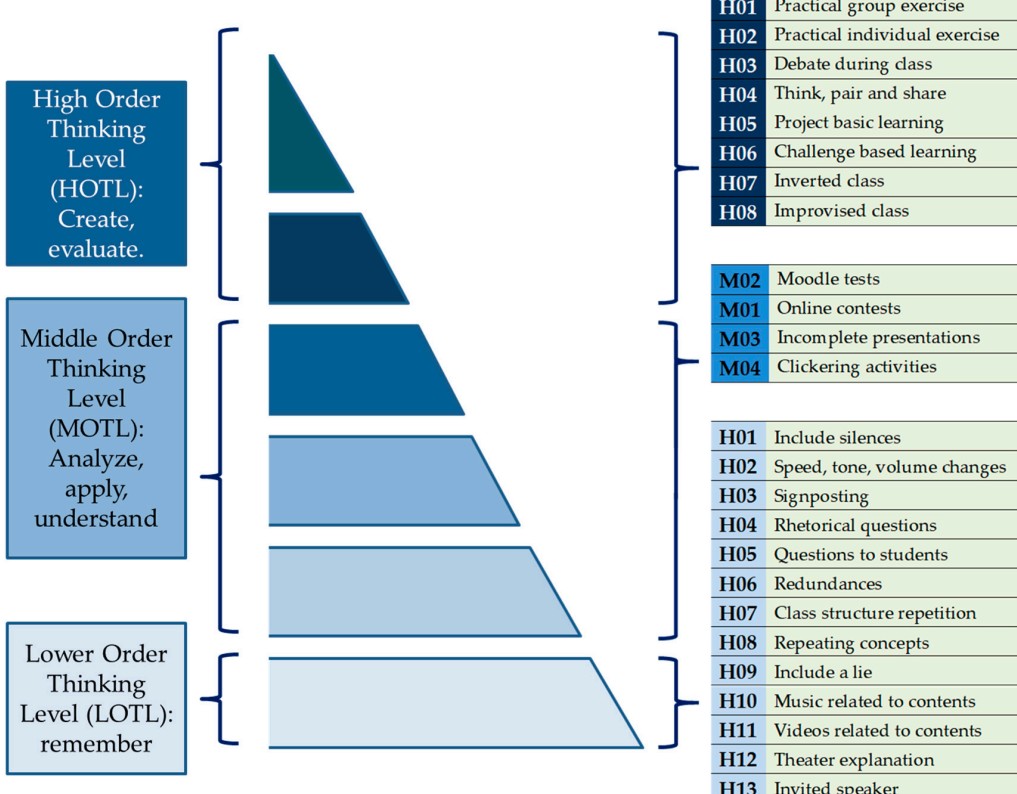

**Figure 2.** Classification of the 25 active alternatives based on Bloom's Taxonomy revised by Anderson.

L01–L08 were the most used LOTL alternatives because they are easy resources for professors to activate students' participation and attention; they include alternatives such as: variations in professors' explanation speed, tone and volume; questions to students and repetitions. M01, Moodle tests about the course contents, was the most used activity to improve MOTL, consisting of tests about the course

contents prepared previously by professors using the Moodle platform so students could do them during the class individually or in groups, with the assistance of the professors and thus receiving immediate results. H02, individual practical exercise, combined with H05, project basic learning (PBL), were the most used alternatives in HOTL, as a type of PBL specifically designed for large groups. They consisted of a brief PBL the students had to solve individually based on the class contents. For instance, they had to solve a construction solution for a specific architectural work in due course by detailing in hand on a small piece of paper like they were an architect on site solving their building team queries in real time.

*4.2. Phase 3: Multi-Criteria Decision Making (MCDM) Model*

As previously stated, in order to be able to integrate different indicators, a MCDM model was chosen because in the current case study, the most sustainable alternative could not be defined by professors directly. This occurred because the considered indicators did not have the same value or tendency for each alternative so that the best alternative according to one issue was not necessarily the best one according to another aspect. In this situation by using a MCDM it was possible to reach an integrated solution that took into account the different issues while, at the same time, it enabled the ranking of these issues according to the case study particularities [62]. To define this new MCDM tool method, the authors reviewed the use of MCDM methodologies for similar research projects. MCDM methodologies applied for university issues have been found in technical literature but they are applications and solve problems which are different from the scope of this project, which proves the novelty of this research project. For example, there are MCDMs to assess learning program quality, teaching quality, learning spaces, curriculum, students' preferences to choose their universities, etc. In this line there is a review on publications about multi-criteria methodologies for university engineering education [63]. This article concludes that until now, most university educational problems studied using MCDM are about resource efficiency (27%), resource location (18%) and assessment (18%). There are numerous MCDM applications to assess sustainability in the field of construction and building technologies [62]. This project used the Integrated Value Model for a Sustainable Evaluation (Modelo Integrado de Valor para una Evaluación Sostenible (MIVES)). The use of MIVES occurred because this methodology allowed researchers to define specific and agile MCDM tools as they have already been successfully carried out in numerous successful research projects [64–68] and this was the type of model needed for this research project. As has been explained in the previous sections, the new tool university professors require should take into account all the specific issues involved in this case study while being easy to use and giving a quick and useful response adaptable to their specific needs and context.

This methodology has the following sub phases: (1.1) establish system limits; (1.2) build the decision making tree composed of requirements, criteria and indicators; (1.3) establish the relative weight each indicator, requirement and criteria has; (1.4) define value functions for each indicator; and (1.5) assess the alternatives. As previously mentioned, these phases relied on technical literature and experts' seminars. This analysis with MIVES was done in three separate groups of alternative active methodologies and, therefore, generated three sustainability rankings: one for higher, another middle and another low order thinking levels. From the different MCDM methods and sustainability assessment tools available, MIVES was chosen because it allowed a complete evaluation, it is agile, can be configured for this case study and has already been combined with other methods and the aforementioned specific mathematical algorithm [69,70].

The boundaries of this MIVES system were based on the initial diagnosis and case study defined in Section 3. Therefore, this new tool evaluated active methodologies and strategies for large group lectures within the "Construction II" course. The implementation of this project used resources and devices currently available in most university classes. These instruments are a computer connected to a projector, a blackboard, Wi-Fi network, a Moodle and at least one computer or smart phone for every two students.

Table 3 presents the decision-making tree for this new MIVES tool, defined by experts in seminars using AHP and relying on extensive literature review as explained in the previous sections. This tree exclusively included the most important and discriminatory indicators [71,72], since for a well-meditated decision, an appropriate requirements tree is of great importance, in which the number of indicators is not excessive. This requirements tree was organized according to the sustainability approach under the three main sustainability requirements: economic, environmental and social [73–75]. As a result of the recommendations provided by expert panels at seminars, the applicability requirement was added.

**Table 3.** Decision making requirements tree with weights in percentage, defined in the experts' seminars.

| Requirements | Criteria | Indicators |
|---|---|---|
| R1. Applicability (25%) | C1. Application (60%) | I01. Easiness to apply (50%) |
| | | I02. Flexibility to adapt (50%) |
| | C2. Transferability (40%) | I03. To other professors (60%) |
| | | I04. To other disciplines (40%) |
| R2. Economic (15%) | C3. Time (100%) | I05. Dedication in class (40%) |
| | | I06. Professor's dedication outside (30%) |
| | | I07. Students' dedication outside (30%) |
| R3. Environmental (10%) | C4. Impact (100%) | I08. Extra environmental impact (100%) |
| R4. Social (50%) | C5. Learning process (Chickering and Gamson principles among others) (45%) | I09. Feedback to students' time (20%) |
| | | I10. Encouraging cooperative work (20%) |
| | | I11. Students and faculty contact (20%) |
| | | I12. Talents and ways of learning (25%) |
| | | I13. Number of concepts (15%) |
| | C6. Innovation (20%) | I14. University learning (55%) |
| | | I15. Large group theoretical classes (45%) |
| | C7. Satisfaction (35%) | I16. Students' (55%) |
| | | I17. Professor's (45%) |

This decision-making tree included the following four requirements, seven criteria and 17 indicators. First, the applicability requirement (R1) had two criteria and four indicators. I01 and I02 assessed two important application aspects: (1) the ease of activities to be prepared, organized, explained and carried out and (2) flexibility of activities to adapt to each class, dedication availability, classroom size and internet connection. I01 did not include application agility because it was included in economic indicators while I02 did not include flexibility to adapt to different professors or disciplines. Indicators I03 and I04 evaluated the transferability strategies have to other professors and other knowledge disciplines respectively. I04 included transference to other schools and faculties, universities, etc.

Second, economic requirement (R2) had three indicators that assessed each professor's and student's dedication during class and outside class. I05 assessed the amount of time spent to carry out each activity and strategy during class time. I06 evaluated each professor's required time in order to prepare and give feedback to students outside of classroom. First year preparation time can be longer than the following years as we can see in Table A2 in Appendix B. This extra time was divided between the different years during which each alternative was expected to be applied. Finally, I07 assessed each student's required time for doing each activity outside the classroom. The required cost to implement these active methodologies and activities was not assessed because, as stated in this research project boundaries, this project considered the available resources. In this sense, this project considered that these new alternatives will use similar materials and resources, such as paper, internet connection, classroom . . . rather than those materials and resources used before these new alternatives developed.

To sum up, the authors of this project considered that during its implementation, there would not be any extra cost or it would be very low and, therefore, no extra cost was taken into account.

Third, I08 unique environmental requirement (R3) indicator assessed the extra impact of each alternative including: extra energy consumption from educational devices such as projectors or clickers etc. and extra waste generation during and outside of class, both by students and professors. This impact was very low compared to the energy consumption from other machines within buildings, such as heating and cooling systems, and the waste generation during other students' and professors' daily activities. However, this requirement and indicator was considered because it contributed to our society environmental impact, increasing crucial parameters such as $CO_2$ emissions. Some activities required additional use of computers that have an extra energy consumption, while other activities used an important amount of extra materials, although these materials were usually paper that can be recycled and recyclable.

Fourth and finally, social requirement (R4) was the most important requirement in this research project that included nine indicators distributed in three criteria: C5 to C7. C5 studied students' learning process assessing: (I09) professor's feedback time to students' queries and exercises, (I10) contribution to teamwork, (I11) promotion of students and professors' relationship and students' sense of belonging to their institution, school, faculty, university [76], (I12) allowing different talents and ways of learning and (I13) number of concepts learned per time unit. It did not include if it promoted deep learning because this new tool was used for the aforementioned three order thinking levels depending on necessities. Nor did this tool incorporate active learning contributions either because the assessed alternatives were active methodologies and strategies. I11 could include students' attitude and interest issues, which in the case study was not crucial because most students were interested but in other courses this aspect could be crucial. This would be assessed in I11. C6 evaluated the contribution each alternative innovation makes to university learning processes in general [77] (I14) and large group theoretical classes specifically (I15). C7 assessed students' satisfaction and their high expectations (I16) as well as individual professor's satisfaction (I17). This satisfaction did not include each alternative feasibility for this case study, which was large groups, because this was also part of the research project boundaries.

These 17 indicators took into account the assessment parameters and the data sources presented in Table 4. These 17 values for each active alternative and strategy are shown in Table A4.

Then, value functions [71] for each 17 indicators were defined based on numerous rigorous bibliographical references which were discussed in the second seminar. All these functions varied between 0 and 1, being 0 the minimum satisfaction and 1 the maximum satisfaction for each indicator, as a response to the 17 indicator values that have different units as described in the previous paragraphs. These adimensional values $V_{I,k}$ could be added and thus the seven criteria satisfaction values $V_{CRi,K}$ were obtained, then the four requirements satisfaction values $S_{IRi,K}$ were obtained and finally the global sustainability index $GS_K$ was obtained. These additions follow Equation (1), Equation (2) and Equation (3) respectively.

$$V_{CRi,k} = \sum_{i=1}^{j} \lambda_{i,k} \cdot V_{i,k}(x_{ind}) \tag{1}$$

$$SI_{Ri,k} = \sum_{i=1}^{j} \lambda_{CRi,k} \cdot V_{CRi,k} \tag{2}$$

$$GS_k = \sum_{i=1}^{j} \lambda_{Ri,k} \cdot SI_{Ri,k} \tag{3}$$

These 17 value functions depended on five parameters, as presented in Equation (4). By giving values to these parameters it was possible to define their shape and, consequently, how each variation of the indicator value was translated into the adimensional scale. For example, if the form was in

*S*, the initial and final variations would have a variation in smaller adimensional values than the central variations.

$$V_{ind} = A + B \cdot \left[1 - e^{-ki \cdot \left(\frac{|Xalt - Xmax|}{Ci}\right)^{Pi}}\right] \tag{4}$$

In Equation (2), A is the value generated by $X_{max}$, the abscissa for the indicator, and *Xalt* is the abscissa for the evaluated indicator that generates a *Vind* value. Pi is a form factor that defines whether the curve is concave, convex, lineal or "S" shaped. *Ci* establishes, in curves with *Pi* > 1, the value of the abscissa at which the inflection point occurs. *Ki* defines the value for the ordinate of point *Ci*. B is the factor for the function to be maintained in the range of 0 to 1 and is defined by Equation (5)

$$B = \left[1 - e^{-ki \cdot \left(\frac{|Xmax - Xmin|}{Ci}\right)^{Pi}}\right]^{-1} \tag{5}$$

Table 5 presents each indicator function shapes, the definition of which relies on previous steps in this research project and experts' seminars. In these seminars it was decided to define linear functions for all indicators for the first applications of this tool. Then, after these initial applications, researchers would decide if it was convenient to define concave functions for the most crucial indicators and convex functions for less important indicators that needed to be promoted.

**Table 4.** Assessment parameters and data sources for each indicator.

|  | **Main Assessed Parameters** | **Data Sources** |
|---|---|---|
| I01 | Requires work before class, during and/or after class | CXP |
| I02 | Adaptable to students', time, space and resources particularities | CXP |
| I03 | Available literature relation to case study and easiness to use | LT |
| I04 | Related 6-digit UNESCO nomenclature areas of expertise | LT |
| I05–7 | Average dedication per class | CXP |
| I08 | Hardware energy consumption and activities waste generation | CXP and LT |
| I09 | Average feedback time | CXP |
| I10 | Encourages cooperative work | CXP and LT |
| I11 | Promotes students and faculty contact | CXP and LT |
| I12 | Allows different styles, approaches, learning and pacing and presentation methods, cultures, recognizes reward and respects creativity | CXP and LT |
| I13 | Average number of concepts | CXP and LT |
| I14 | University previous projects and literature about this alternative | LT |
| I15 | Large groups previous projects and literature about this alternative | LT |
| I16 | Satisfaction questionnaires | CXP |
| I17 | Focus groups about satisfaction | CXP |

Legend: CXP—2017 to 2018 course experience; LT—literature review.

**Table 5.** New tool indicators functions shapes.

| Indicators Code | Unit | Function Shape | X min. | X max. | C | K | P |
|---|---|---|---|---|---|---|---|
| I01, I02, I03, I04, I10, I11, I12, I14, I15, I16, I17 | Points | IL |  | 100 | 50 |  |  |
| I05 | Minutes | DL |  | 180 | 90 |  |  |
| I06 | Minutes | DL | 0 | 60 | 30 | 0.5 | 1.25 |
| I07 | Minutes | DL |  | 90 | 45 |  |  |
| I08 | Points | DL |  | 100 | 50 |  |  |
| I09 | Hours | DL |  | 168 | 84 |  |  |
| I13 | Concepts/hour | IL |  | 25 | 12 |  |  |

Legend: IL means increasing lineal, and DL means decreasing lineal.

### 4.3. Mathematic Algorithm

The Knapsack algorithm was chosen for phase five because it was able to generate sets of active methodologies and strategies designed for specific cases of classes and took into account the contents of

each class, its order thinking level and the 17 chosen indicators. Moreover, Knapsack has already been successfully used to do so in combination with MIVES as previously stated. This algorithm defines sets of alternatives, maximizing some values according to the required measures [78]. Knapsack results are one or more sets that comply with the total measurement options equal to or less than the required measure and with the maximum satisfaction for the chosen value. In this research project the measure was the time available for the active methodologies in the class and the value was the sustainability index $GS_K$ of the alternatives, explained in detail in Section 4.2.

So that the Knapsack algorithm identified optimized sets of alternatives, this algorithm was introduced in C++ software using dynamic programming to reduce operation time. Equation (4) presents the Knapsack problem for this research, in which $GS_K$ is the value required to be maximized. The constraints for this problem were the minimum and maximum class time availability for active methodologies and strategies (W1, W2), the integers were the time that each active methodology requires (Tn) and the sum of integers was the total time spent with active methodologies [78,79]. These availability times per session are presented in Table A2 and each active strategy time are shown as indicator I05, in Table A4.

Knapsack was run for each of the 15 different course classes and for each type of OTL cases: L, L and M, L and M and H and M and H. For classes that had exactly the same availability time and OTL case more than one option was given by running Knapsack twice, discarding the first resulting set (Kn1) in the second run (Kn2).

$$W_1 \leq \sum_1^i T_n \leq W_2 \text{ Maximise } \frac{\sum_1^i SI_n * T_n}{\sum_1^i T_n} \tag{6}$$

$T_n$: Time that each active methodology/technology "$n$" requires; $W_1$, $W_2$: minimum and maximum class time availability for active methodologies and strategies; $i$: number of items in subset; $SI_n$: sustainability index of site $n$.

## 5. Results

The main results of this project are: the alternatives sustainability indexes from Phase 3 and the sets of alternatives from Phase 5, as explained in Section 4 and Figure 1. The alternatives sustainability indexes are presented in Table 6, with a ranking for each order thinking level.

**Table 6.** Alternatives requirements satisfaction values and global sustainability index.

| Alternative | $SI_{R1,K}$ | $SI_{R2,K}$ | $SI_{R3,K}$ | $SI_{R4,K}$ | $GS_K$ | OTL | Ranking |
|---|---|---|---|---|---|---|---|
| L01–L08 | 0.73 | 0.98 | 1.00 | 0.57 | 0.72 | | 2 |
| L09 | 0.62 | 0.93 | 0.96 | 0.60 | 0.69 | | 4 |
| L10 | 0.70 | 0.94 | 1.00 | 0.52 | 0.68 | LOTL | 5 |
| L11 | 0.88 | 0.85 | 1.00 | 0.52 | 0.71 | | 3 |
| L12 | 0.67 | 0.91 | 1.00 | 0.56 | 0.68 | | 5 |
| L13 | 0.66 | 0.90 | 1.00 | 0.66 | 0.73 | | 1 |
| M01 | 0.60 | 0.83 | 0.96 | 0.57 | 0.66 | | 2 |
| M02 | 0.58 | 0.85 | 0.96 | 0.65 | 0.69 | MOTL | 1 |
| M03 | 0.63 | 0.90 | 0.05 | 0.53 | 0.56 | | 3 |
| M04 + H04 | 0.66 | 0.90 | 0.92 | 0.67 | 0.73 | | 2 |
| H01 | 0.63 | 0.73 | 0.95 | 0.69 | 0.71 | | 3 |
| H02 + H05 | 0.73 | 0.73 | 0.92 | 0.57 | 0.67 | | 4 |
| H03 | 0.66 | 0.90 | 0.92 | 0.63 | 0.71 | HOTL | 3 |
| H06 | 0.56 | 0.84 | 0.92 | 0.84 | 0.78 | | 1 |
| H07 | 0.67 | 0.52 | 0.92 | 0.46 | 0.57 | | 5 |
| H08 | 0.58 | 0.70 | 1.00 | 0.71 | 0.71 | | 3 |
| Average | 0.66 | 0.84 | 0.90 | 0.61 | 0.69 | | |

Legend: OTL—order thinking level; $SI_{Ri,K}$—requirements satisfaction values; $GS_K$—global sustainability index.

The Knapsack algorithm results are presented in Table 7. These results include the most sustainable sets of alternatives, times and GSK considering both aforementioned cases Kn1 and Kn2.

**Table 7.** Knapsack results in first run (Kn1) and also second run (Kn2) in classes with exactly the same OTL and time availability.

| Contents, Topic | Knapsack Result Kn1 | | | Knapsack Result Kn2 | | |
|---|---|---|---|---|---|---|
| | Activities | Time | GS$_K$ | Activities | Time | GS$_K$ |
| 1. Introduction | L01–08, L10, L13 | 50 | 0.72 | – | – | – |
| 2. Site soil | L13 | 30 | 0.73 | – | – | – |
| 3. Geotechnical | L01–08, L10–13 | 80 | 0.71 | – | – | – |
| 4. Retaining wall | L01–08, L11, L13, M01 | 80 | 0.71 | – | – | – |
| 5. Diaphragm walls | L13 | 30 | 0.73 | L01–08, L11 | 30 | 0.72 |
| 6. Foundations criteria | M04 + H04 | 30 | 0.73 | – | – | – |
| 7. Foundations types | L01–08, L11, L13 | 60 | 0.72 | – | – | – |
| 8. Slab floors | L01–08, L10, L13 | 50 | 0.72 | L11, M02 | 45 | 0.70 |
| 9. Timber structures | L01–08, L11, L13 | 60 | 0.72 | L09, M02 | 60 | 0.69 |
| 10. Concrete block | L13 | 30 | 0.73 | – | – | – |
| 11. Steel structures | L13 | 30 | 0.73 | M02 | 30 | 0.69 |
| 12. Brick walls | L13 | 30 | 0.73 | M04 + H04 | 30 | 0.73 |
| 13. RC criteria | H06 | 60 | 0.78 | – | – | – |
| 14. RC types | M01, M04 + H04 | 50 | 0.70 | – | – | – |
| 15. Precast concrete | M04 + H04, H01, H03, H06 | 150 | 0.74 | – | – | – |

## 6. Discussion

The previous results prove that the application of this new tool based on MIVES methodology and the Greedy–Knapsack algorithm was successful and we obtained satisfactory values and sustainability indexes for the 15 alternatives (see Table 6) and the best sets of alternatives (see Table 7) for each course session, considering exactly the available time and an increase of 20% of this time. From the global sustainability indexes in Table 6 we observed that for this specific case study all alternatives have similar satisfactory sustainability indexes, from 0.57 to 0.78, with an average value of 0.69. From these indexes we could conclude that all alternatives are satisfactory, but they have room for improvement. Analysing the four requirements satisfaction values average (Table 6) we can conclude that these alternatives weakest points are their applicability and social indicators. Therefore, in order to improve these alternatives in this study case, the most effective action would be improving their application and learning processes.

The most sustainable alternative is "H06. Challenge Based Learning" while the least sustainable is "H07. Inverted Class". These indexes respond to this specific case study and, therefore, for different case studies and study boundaries, these indexes could be different. The average sustainability index for each alternatives group regarding their order thinking level was 0.70, 0.66 and 0.69 for LOTL, MOTL and HOTL respectively. These average indexes are very similar with a major difference in the case of MOTL.

Tables 8 and 9 compare the Knapsack proposals with the professors' proposals before considering Knapsack presented in Table A2. They compare time and global sustainability index GS$_K$ for each case respectively.

Tables 8 and 9 mainly show differences between professors' initial proposal (Pri) and Knapsack proposals (Kn1 and Kn2). These differences are obviously due to the fact that both proposals have completely different approaches, limits and potentials. Pri was previous to this new tool definition and application, so it was defined manually by the course teaching team which was able to consider very important facts derived from their skills, experience and expertise as well as intuition and adaptability to unexpected variables among others. For example, Pri included the lecturer's abilities and willingness to carry out each of the alternatives, the keynote speaker's specialty and availability in alternative L13, or previous real experiences from performing each of the active teaching alternatives. On the other hand, Knapsack proposals were able to integrate all the important indicators presented in the

requirements tree (Table 3) taking into account the MIVES methodology and the decisions taken by multidisciplinary experts in the seminars. In consequence, their sustainability index satisfaction was lower as presented in Table 9. At the same time, these proposals fit exactly with the available time that had been foreseen by the teaching team to dedicate to them.

**Table 8.** Time of professors' proposal before defining Knapsack and Knapsack proposal times.

| Topic | Time Variation | | | Activities Coincidence | | |
|---|---|---|---|---|---|---|
| | **Pri** | **Kn1** | **Kn2** | **Pri–Kn1** | **Pri–Kn2** | **Kn1–Kn2** |
| 1 | 10.00% | 0.00% | – | 33.33% | – | – |
| 2 | 17.00% | 0.00% | – | 0.00% | – | – |
| 3 | 13.00% | 0.00% | – | 29.17% | – | – |
| 4 | 19.00% | 0.00% | – | 50.00% | – | – |
| 5 | 17.00% | 0.00% | 0.00% | 0.00% | 50.00% | 0.00% |
| 6 | 0.00% | 0.00% | – | 100.00% | – | – |
| 7 | 0.00% | 0.00% | – | 66.66% | – | – |
| 8 | 0.00% | 0.00% | −10.00% | 33.33% | 41.67% | 0.00% |
| 9 | 0.00% | 0.00% | 0.00% | 33.33% | 0.00% | 0.00% |
| 10 | 17.00% | 0.00% | – | 0.00% | – | – |
| 11 | 0.00% | 0.00% | 0.00% | 0.00% | 0.00% | 0.00% |
| 12 | 17.00% | 0.00% | 0.00% | 0.00% | 0.00% | 0.00% |
| 13 | 0.00% | 0.00% | – | 0.00% | – | – |
| 14 | 0.00% | 0.00% | – | 50.00% | – | – |
| 15 | 13.00% | 0.00% | – | 26.50% | – | – |
| | | | Average | 28% | 18% | 0% |
| | | | Standard deviation | 0.30 | 0.25 | 0.00 |

**Table 9.** $GS_K$ professors' proposal before defining Knapsack and Knapsack proposal.

| Topic | $GS_K$ | | | Activities Coincidence | | |
|---|---|---|---|---|---|---|
| | **Pri** | **Kn1** | **Kn2** | **Pri–Kn1** | **Pri–Kn2** | **Kn1–Kn2** |
| 1 | 0.68 | 0.72 | – | −5% | – | – |
| 2 | 0.53 | 0.73 | – | −20% | – | – |
| 3 | 0.71 | 0.71 | – | −1% | – | – |
| 4 | 0.65 | 0.71 | – | −6% | – | – |
| 5 | 0.53 | 0.73 | 0.72 | −20% | −18% | 2% |
| 6 | 0.73 | 0.73 | – | 0% | – | – |
| 7 | 0.64 | 0.72 | – | −9% | – | – |
| 8 | 0.69 | 0.72 | 0.70 | −3% | 0% | 3% |
| 9 | 0.63 | 0.72 | 0.69 | −9% | −6% | 3% |
| 10 | 0.71 | 0.73 | – | −2% | – | – |
| 11 | 0.72 | 0.73 | 0.69 | −2% | 2% | 4% |
| 12 | 0.71 | 0.73 | 0.73 | −2% | −2% | 0% |
| 13 | 0.62 | 0.78 | – | −17% | – | – |
| 14 | 0.67 | 0.70 | – | −4% | – | – |
| 15 | 0.70 | 0.74 | – | −4% | – | – |
| | | | Average | −7% | −5% | 2% |
| | | | Standard deviation | 0.30 | 0.07 | 0.08 |

At the same time, Pri was not strict with time and, in consequence, the spent time was greater than that expected, and this fact brought dissatisfaction from both students and professors because some classes were too full of activities. While Kn1 and Kn2 were not able to consider other variables beyond the indicators incorporated in Table 3 and/or those considered in the seminar, indicators such as: unexpected circumstances, suddenly on-time losses or any specific lecturer skill and experience. To sum up and as previously said, these proposals denote two main strengths for each approach: (1) professors' and teaching team's potentials in bringing experience, skills, expertise and rapid response and (2) the power the new tool provides in assisting professors to plan and manage their lectures while incorporating active learning methodologies.

These two strengths of the research project's new tool were found in the seminars and focus groups experts participated in while discussing the previously presented results. From the application point of view, the experts highlighted the strength this new tool has to help unexperienced professors when they start giving lectures and assist busy senior professors who are willing and aware of the importance to incorporate active methodologies in their lectures. The experts also pointed out the aforementioned limitations based on the result of this new tool, which make necessary the interpretation of these results by the professors before applying them. In consequence, Kn1 and Kn2 are the best results as advice to professors, who should complement, improve and adapt them taking into account their experience, skills, specific context and any on-time change. In this sense, the results of applying this tool should not be considered as a completely finished and closed result to be applied without more considerations.

In these seminars, experts detected several potential future improvements, which would mean defining a perfected version of this new tool that will: (a) be more interactive with professors who will be able to modify customizable values and give feedback for the results obtained in order to keep improving the tool; (b) have an easier and more friendly interface such as becoming an App; (c) suggest sets of activities for a whole class, for a group of classes or for the whole course, multiplying this new tool potential to assist professors in managing their teaching activities; (d) include crucial neuroscience aspects [80], such as improving learning processes by activating the emotional part of students' brains; (e) add a more complete database of active learning alternatives for lectures and their features in detail; (f) be ready to be applied to other areas within architecture studies and beyond other disciplines and (g) incorporate artificial technologies such as artificial intelligence [81] in order to include aspects not now considered, such as professors' experience.

## 7. Conclusions

The main novelties of this research project are the successful definition and first application of an innovative new multi-criteria decision making tool that assists professors to find the best set of active learning methodologies to be applied in a lecture, while taking into account multiple indicators and the order thinking level of each alternative as well as the class contents. This new tool has been defined following MIVES and incorporates Knapsack. This model is based on seminars where experts participated and used value functions to integrate all the different indicators considered. The use of this methodology relies upon previous successful applications at academic and professional levels in other fields of expertise.

This new tool has been defined to contribute towards solving a current general problem which university lectures pose, starting with its application in the specific discipline of Architectural Technologies and specifically at the Barcelona School of Architecture. This current tool has a strong potential to assist professors while incorporating active learning methodologies in their lectures. Moreover, it promotes professors' management of their courses in terms of having greater control over their class time and their course contents order thinking levels, among other benefits. There are different previous experiences that introduced Bloom's Taxonomy obtaining a similar positive result [72]. This tool also has limitations that result in the need for a manual application of these results.

In the future, improved versions of this tool could minimize or even overcome these limitations by increasing its interactivity with users, opening their scope, incorporating a more complete database, neuroscience issues and artificial technologies, as has been explained in the previous section. These future actions are expected to be part of a broader research project in which more researchers would participate in order to continue to increasingly help professors in their teaching activities in order to improve students' learning processes.

**Author Contributions:** O.P. lead this research project and methodology conceptualization helped by J.F. and S.M.A.H.; investigation and writing, O.P; analysis, O.P and S.M.A.H.; supervision, J.F.

**Funding:** This research received no external funding.

**Acknowledgments:** The authors are grateful to Daniel García from UPC, ETSAB member of Gilda; to Josep Maria Gonzàlez from UPC, ETSAB, TA; to S.M. Amin Hosseini from UPC, DECA; to Gemma Muñoz from UPC,

ETSAB, TA + La Salle, to Marta Cabré from PSITIC, to Rosana Fernández and Silvia Lope from CESIRE, to Antoni Perez, Noelia Olmedo and Mireia Lopez from ICE arquitectura and ETSAB students Ainhoa Varela and Marcel Xalabarder. The author, Oriol Pons Valladares, is a Serra Húnter Fellow.

**Conflicts of Interest:** The authors declare no conflict of interest.

## Appendix A

**Table A1.** The abbreviations used in the text.

| Abbreviations | Relevant Values |
|---|---|
| AHP | Analytic hierarchy process |
| MCDM | Multi-criteria decision making |
| LOTL | Lower order thinking level |
| MOTL | Middle order thinking level |
| HOTL | High order thinking level |
| MIVES | Modelo Integrado de Valor para una Evaluación Sostenible (Integrated Value Model for a Sustainable Evaluation) |
| CXP | 2017 to 2018 course experience |
| LT | Literature review |
| PBL | Project basic learning |
| $V_{I,k}$ | Indicators satisfaction values |
| $V_{CRi,K}$ | Criteria satisfaction values |
| $SI_{Ri,K}$ | Requirements satisfaction values |
| $GS_K$ | Global sustainability index |
| IL | Increasing lineal |
| DL | Decreasing lineal |
| Pri | Professors initial proposal of activities per class |
| Kn1 | Knapsack first run proposal of activities per class |
| Kn2 | Knapsack second run proposal of activities per class |

## Appendix B

**Table A2.** "Construction II" main contents classification in the three order thinking levels (OTL): lower order thinking level (LOTL), middle order thinking level (MOTL), high order thinking level (HOTL).

| Contents, Topic | Main Contents | Duration | | | OTL | Professors' Previous Initial Proposal | |
|---|---|---|---|---|---|---|---|
| | | Cl | Ex | Act | | Alter | Time |
| 1. Introduction. Soil | Introduction and summary. Soil identification and values | 150 | 100 | 50 | LOTL MOTL | L01–L08, 2*M01 | 55 |
| 2. Site soil | Site soil cases | 120 | 90 | 30 | LOTL MOTL | L01–L08, M01 | 35 |
| 3. Geotechnical report | Definition and contents | 180 | 100 | 80 | LOTL | L01–L08, L09, L12, L13 | 90 |
| 4. Retaining wall | Definition, types, design and construction process | 180 | 100 | 80 | LOTL MOTL | L11, M01, M02, M03 | 95 |
| 5. Diaphragm walls | Definition, types, design and construction process | 180 | 150 | 30 | LOTL MOTL | L01–L08, M01 | 35 |
| 6. Foundations criteria | Criteria to choose types of foundations | 180 | 150 | 30 | MOTL HOTL | M04 + H04 | 30 |
| 7. Foundations types | Foundations types and applications | 180 | 120 | 60 | LOTL MOTL | L01–L08, L11, M03 | 60 |
| 8. Slab floors | Definition and types. Design and construction process | 180 | 130 | 50 | LOTL MOTL | L01–L08, L11, M01 | 50 |
| 9. Timber structures | Definition, types, design and construction process | 120 | 60 | 60 | LOTL MOTL | L01–L08, L12, M03 | 60 |

**Table A2.** *Cont.*

| Contents, Topic | Main Contents | Duration | | | OTL | Professors' Previous Initial Proposal | |
| --- | --- | --- | --- | --- | --- | --- | --- |
| | | Cl | Ex | Act | | Alter | Time |
| 10. Concrete block walls structures | Design, construction process, types, application criteria and examples | 120 | 90 | 30 | LOTL MOTL HOTL | L10, H01 | 35 |
| 11. Steel structures | Definition, types, design, building process, examples | 180 | 150 | 30 | LOTL MOTL | L01–L08, L11 | 30 |
| 12. Brick walls structures | Design, construction process, types, application criteria and examples | 180 | 150 | 30 | LOTL MOTL HOTL | L10, H03 | 35 |
| 13. Reinforced concrete (RC) criteria | Criteria to choose RC structure type | 180 | 120 | 60 | MOTL HOTL | M03, H02 + H05 | 60 |
| 14. RC types | Design, construction process, types, application criteria and examples | 180 | 130 | 50 | MOTL HOTL | M01, H02 + H05 | 50 |
| 15. Precast concrete | Design, construction process, types, application criteria and examples | 180 | 30 | 150 | MOTL HOTL | M01, H06, H08 | 170 |

Legend: Cl—Duration of the class, Ex—Duration of the explanation, Act—Duration of the active strategies or methodologies, Alter—Alternatives proposed in professors' first initial proposals previous to Knapsack.

**Table A3.** Alternatives for the three order thinking levels.

| Code | Alternative | Sessions 2017–2018 |
| --- | --- | --- |
| H01 | Practical exercise in groups during the class related to the contents | 6, 8/2/2018 |
| H02 | Individual practical exercise during the class related to the contents | 15/2,19/4 |
| H03 | Debate during class. It requires preparation | 10,15/5/2018 |
| H04 | Think, pair and share: activity that involves thinking individually, exchanging information between two and sharing to a bigger group | 15/2, 22/3 |
| H05 | Project basic learning (PBL) designed and prepared for large groups | 15/2,19/4 |
| H06 | Challenge based learning designed and prepared for large groups | 10,15/5/2018 |
| H07 | Inverted class available in audio or video on the Moodle platform before the real class, which is dedicated to solving queries and practical exercises | 5/2/2018 |
| H08 | Improvised class based on students' queries and needs while covering the planned contents | 10,15/5/2018 |
| M01 | Moodle tests about the course contents done by students during the class | All sessions |
| M02 | Online contests using digital platforms, also about course contents | 6, 8/2/2018 |
| M03 | Incomplete presentations for students to complete during class | 29/1-6/3/2018 |
| M04 | Tests or PBL using clickers technology or raising hands | 15/2, 22/3 |
| L01 | Include silences in between professor's explanation to let and promote students thinking | |
| L02 | Speed, tone, volume variation of professor's explanation | |
| L03 | Include signposting to help students to connect different concepts | |
| L04 | Include rhetorical questions to help students to keep the attention | All sessions |
| L05 | Include questions during the explanation and let students answer | |
| L06 | Use redundancies and repetitions of the most important concepts | |
| L07 | Class structure repetition to help students following classes | |
| L08 | Repeat concepts during class introduction, body and ending | |
| L09 | Include a lie in the professors' discourse that students have to find | 3, 8/5/2018 |
| L10 | Music related to the class contents at the beginning of the class | |
| L11 | Videos from internet platforms related to class contents | All sessions |
| L12 | Theatre explanation related to class contents | 22/2, 1/3/2018 |
| L13 | Invited speaker or expert explains a specific related topic | 20/3, 8/5/2018 |

These activities were applied during the previous course 2017–2018. The following alternatives classified as LOTL can be used for the understanding MOTL: L11. Related videos, L12. Theatre explanation, L13. Invited speaker.

## Appendix C

**Table A4.** I1–I17 indicator values for each active methodology and strategy alternative.

| Alter. | Assessed Indicators | | | | | | | | | | | | | | | | |
|---|---|---|---|---|---|---|---|---|---|---|---|---|---|---|---|---|---|
| | I01 | I02 | I03 | I04 | I05 | I06 | I07 | I08 | I09 | I10 | I11 | I12 | I13 | I14 | I15 | I16 | I17 |
| L01–L08 | 93 | 80 | 13 | 100 | 15 | 0 | 0 | 0 | 0 | 0 | 0 | 33 | 20 | 93 | 0 | 77 | 70 |
| L09 | 80 | 70 | 0 | 82 | 30 | 5 | 0 | 6 | 3 | 0 | 0 | 33 | 20 | 100 | 100 | 66 | 60 |
| L10 | 80 | 90 | 2 | 100 | 5 | 14 | 0 | 0 | 0 | 0 | 25 | 22 | 12 | 98 | 0 | 53 | 60 |
| L11 | 70 | 90 | 95 | 81 | 15 | 30 | 0 | 0 | 0 | 0 | 25 | 33 | 12 | 32 | 0 | 84 | 90 |
| L12 | 60 | 70 | 43 | 71 | 15 | 18 | 0 | 0 | 0 | 0 | 25 | 33 | 8 | 94 | 0 | 75 | 90 |
| L13 | 80 | 60 | 16 | 100 | 30 | 15 | 0 | 0 | 0 | 0 | 50 | 56 | 10 | 96 | 0 | 82 | 90 |
| M01 | 85 | 50 | 4 | 97 | 20 | 24 | 15 | 6 | 0 | 0 | 0 | 33 | 24 | 96 | 0 | 80 | 70 |
| M02 | 65 | 50 | 16 | 97 | 30 | 24 | 0 | 6 | 0 | 100 | 25 | 22 | 20 | 93 | 0 | 71 | 80 |
| M03 | 90 | 70 | 11 | 54 | 30 | 13 | 0 | 94 | 0 | 0 | 25 | 33 | 14 | 98 | 0 | 51 | 80 |
| M04 + H04 | 80 | 50 | 28 | 100 | 30 | 15 | 0 | 12 | 72 | 100 | 50 | 100 | 6 | 68 | 6 | 73 | 70 |
| H01 | 80 | 40 | 33 | 89 | 30 | 45 | 0 | 8 | 72 | 100 | 50 | 78 | 6 | 83 | 0 | 71 | 70 |
| H02 + H05 | 70 | 40 | 79 | 100 | 30 | 45 | 0 | 12 | 72 | 0 | 25 | 78 | 6 | 73 | 0 | 71 | 60 |
| H03 | 48 | 40 | 81 | 93 | 30 | 15 | 0 | 12 | 108 | 100 | 75 | 100 | 6 | 8 | 0 | 78 | 70 |
| H06 | 70 | 50 | 3 | 100 | 60 | 15 | 0 | 12 | 108 | 100 | 100 | 100 | 3 | 98 | 78 | 78 | 80 |
| H07 | 70 | 50 | 41 | 99 | 60 | 18 | 90 | 12 | 0 | 0 | 25 | 33 | 20 | 34 | 0 | 55 | 70 |
| H08 | 80 | 50 | 4 | 88 | 90 | 30 | 0 | 0 | 0 | 100 | 50 | 67 | 13 | 99 | 0 | 77 | 80 |

Legend: Alter.—Alternatives.

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
