# Peer review of "Integrated Value Model to Assess the Sustainability of Active Learning Activities and Strategies in Architecture Lectures for Large Groups"

_sustainability, doi:10.3390/su11102917_

Round 1
Reviewer 1 Report
This paper developed and applied a new MCMD tool incorporating a mathematical algorithm in order to choose the best set of active learning activities for university students, starting with its application in the specific discipline of Architectural Technologies.
Present research includes novelty scientific developments, as required for a good quality journal, by using a systematic and wellstructured methodology.
Authors have applied the MIVES methodology (long time approached in several research fields) in a set of 15 alternatives by using 17 indicators in teaching the best practices in academic courses on architecture. Additionally, these researchers discuss their results in a deep detail with interesting concluding remarks.
Abstract
It is welcome that authors underline, firstly, how this research paper focuses on architecture schools and departments. In addition, they should clearly state the main outputs from the achieved work.
Subsection 4.2
Line 184, authors said: “…have already been successfully carried out in numerous successful research projects [64] ….” This is only one self‐cited reference [64]. Therefore, it seems adequate opening references to other MIVES developers, as listed below:
‐ J. Cuadrado, E. Rojí, J.T. San‐José, J.P. Reyes (2012). Sustainability index for industrial buildings. Proceedings of the Institution of Civil Engineering ‐ structures and buildings 164(SB5):245‐253.
‐ M. Zubizarreta, J. Cuadrado, J. Iradi, H. García, A. Orbe (2017). Innovation evaluation model for macro‐construction sector companies: A study in Spain. Evaluation and Program Planning 61:22‐37.
‐ A. Aguado, A. del Caño, P. de la Cruz, P.J. Gómez, A. Josa (2012). Sustainability Assessment of Concrete Structures within the Spanish Structural Concrete Code. Journal of Construction Engineering and Management ASCE 138 (2): 268‐76.
‐ J.T. San‐José, I. Garrucho, J. Cuadrado (2006). The first sustainable industrial building projects. Proceedings of the Institution of Civil Engineering ‐ Municipal Engineer 159(3):147‐153.
Lines 197‐198, it is remarked the same comment because only a new self‐citation is appended, Nr. 65.
Section 4. Discussion
Lines 386‐417. Because this part is out of the scope of present work (future researches); please, condensate.
Section 5. Conclusions
Lines 435‐441. The same comment as in section 4.
Erratum:
‐ Not referenced in the text the Table A.1.
Author Response
Dear Sir/Madam,
First of all we would like to thank you for your review, which has been really useful to improve our research paper.
To facilitate the reviewing process we have written in red all the changes in the manuscript and in the caption figures.
- In response to your comment about improvements in the abstract we have added:
At present university professors lack tools to know which is the most sustainable activity and/or strategy to incorporate into our large groups theoretical classes in order to improve our students’ learning process taking into account each scenario. These scenarios have different order thinking levels, numbers of students, available time, classroom size and professor skills, among other factors to consider. In architecture schools we have this problem in theoretical lectures. This project has developed and applied a new Multi Criteria Decision Making tool incorporating a mathematical algorithm in order to choose the best set of active learning activities for each case for these lectures in architectural technology courses. This process has relied on Seminars involving experts and the use of The Integrated Value Model for Sustainable Assessment. This tool has been very useful to solve the aforementioned problems because architecture professors have been able to choose the most sustainable activity for each scenario considering the alternative sustainability indexes. This first application has been highly useful to assist professors to incorporate active learning methodologies in their classes and to promote lecturers’ management of their courses contents and time. Future improved versions of this tool will increase its interactivity and broaden its scope.
- In response to your comments about Subsection 4.2, we have added the suggested references among others in line 214 “[65]–[69]” and in line 202 “with other methods and the aforementioned specific mathematical algorithm [70], [71].”.
- In response to your comment about Discussion we have changed this part to:
In these seminars, experts detected several potential future improvements, which would mean defining a perfected version of this new tool that will: a) be more interactive with professors who will be able to modify customizable values and give feedback for the results obtained in order to keep improving the tool; b) have an easier and more friendly interface such as becoming an App; c) suggest sets of activities for a whole class, for a group of classes or for the whole course, multiplying this new tool potential to assist professors to manage their teaching activities; d) include crucial neuroscience aspects [81], such as improving learning processes by activating the emotional part of students’ brains; e) add a more complete database of active learning alternatives for lectures and their features in detail; f) be ready to be applied to other areas within architecture studies and beyond other disciplines and g) incorporate artificial technologies such as Artificial Intelligence [82] in order to include aspects not now considered, such as professors’ experience.
- In response to your comment about Conclusions
In the future, improved versions of this tool could minimize or even overcome these limitations by increasing its interactivity with users, opening their scope, incorporating a more complete database, neuroscience issues and artificial technologies, as has been explained in the previous section.
- In response to your comment about Table A.1., in line 131 we have added “Appendix A presents a complete list of abbreviations”.
Reviewer 2 Report
The topic is extremely interesting since purely goodwill trial and error dynamics in teaching are risky and unnecessary in times when this kind of assessment methods are being created and developed. However it is rather curious that the paper considers much more important the assessment method for the new teaching methodologies than the methodologies themselves.
Thus, the correctness of the assessment of these new methodologies under discussions is rather difficult to follow when first comments about their nature are not provides until page 11 (lines 341 and 342) and a very good detailed list is not provided until page 16 within an annex which is not referred in the main text until page 9 (line 311).
That fact produces a general abstraction which is increased by the abundance of unusual acronyms and abbreviations employed.
However, the paper itself provides good clues about how to amend this lightly uncomfortable abstraction. Its first pages frame the course issues in a smart manner talking about the context of the course, the participants of the experiment and its goals. If these introductory descriptions were followed by a more detailed description of the problematics of the course, the origin, and classification of the new intended methodologies, and not just by numerical results of their application, the whole paper would be more lively and understandable.
Line 530, “falcon” should be written with capital letter, so “Falcon”.
Author Response
Dear Sir/Madam,
First of all we would like to thank you for your review, which has been really useful to improve our research paper.
To facilitate the reviewing process we have written in red all the changes in the manuscript and in the caption figures.
- In response to your general comment about this paper focusing on these methodologies sustainability assessment rather than in the case study and the methodologies used description we recognize that this is the case. We understand the reviewer interesting and important comment but we have focused on this assessment because this is this paper main novelty. The case study description is the main issue of a previous paper and these methodologies description has already been done in numerous previous technical literature. The authors, due to this scientific paper extension limitation have decided to refer to the aforementioned previous research projects. This is the reason why these methodologies description is referred to other papers that describe these methodologies in detail such as the methodologies reviewed in section 2 and the previous article from the authors, references 60 and 61. In other words, our novel strategy, sustainability method, is capable to be applied for other teaching & learning methods as well. However, in this research study, the novel strategy has been designed especially for these specific teaching & learning method, which have already been explained in 60 and 61. Nevertheless, to improve this lack according the reviewer important comment we have added explanations in table B.2. and we have added in section 4.1:
These first phases are presented in Figure 2 and in Annex B. Table B.1 summarizes the course contents and Table B.2 lists the 25 alternatives.
L01-L08 were the most used LOTL alternatives because they are easy resources for professors to activate students’ participation and attention; they include alternatives such as: variations in professors’ explanation speed, tone and volume; questions to students and repetitions. M01, Moodle tests about the course contents, was the most used activity to improve MOTL, consisting of tests about the course contents prepared previously by professors using the Moodle platform so students could do them during the class individually or in groups, with the assistance of the professors and thus receiving immediate results. H02, individual practical exercise, combined with H05, project basic learning (PBL), were the most used alternatives in HOTL, as a type of PBL specifically designed for large groups. They consisted of a brief PBL the students had to solve individually based on the class contents. For instance, they had to solve a construction solution for a specific architectural work in due course by detailing in hand on a small piece of paper like they were an architect on site solving their building team queries in real time.
- Similarly, in section 3.1 we describe the main issues about the course problems, their origin and the new intended methodologies. Because they are explained in detail in references 60 and 61. Nevertheless, in order to clarify it we have added “as it is explained in detail by Pons et al. [60]” in line 101 and “as presented in depth by Pons & Franquesa [61]” in line 103. To minimize the abstraction we have added Figure 2.
- In response to your final minor comment, in Line 530, “falcon” has been written with capital letter, so “Falcon”.
Reviewer 3 Report
The paper describes a new means for determining the best teaching strategy for a large architectural course. While this is definitely an interesting subject, the paper claims that the method developed can be universally applied to any large groups' lecture. I doubt this claim, maybe the focus on a specific course environment should be made clearer.
The paper is pleasant to read and presents method and results in a comprehensible manner. Some minor remarks: For me, the genesis of the requirements tree in Table 3 is not understandable, the cited sources 66 and 67 don't provide much help. Also, the most important alternatives of Table B.1 should be explained better and not only in the Appendix.
Author Response
Dear Sir/Madam,
First of all we would like to thank you for your review, which has been really useful to improve our research paper.
To facilitate the reviewing process we have written in red all the changes in the manuscript and in the caption figures.
- In response to your first comment we have clarified more in the title, abstract and introduction (line 49) that this first model and application have been in architecture studies and the Architectural Technology course.
- In response to your first minor remark we have added “defined by experts in seminars using AHP and relying on extensive literature review as explained in the previous sections.” in the text and “defined in the experts’ seminars.” in table 3 caption.
- In response to your second minor remark we have added new references.
- In response to your third minor remark we have added explanations in table B.2. and we have added in section 4.1:
These first phases are presented in Figure 2 and in Annex B. Table B.1 summarizes the course contents and Table B.2 lists the 25 alternatives.
L01-L08 were the most used LOTL alternatives because they are easy resources for professors to activate students’ participation and attention; they include alternatives such as: variations in professors’ explanation speed, tone and volume; questions to students and repetitions. M01, Moodle tests about the course contents, was the most used activity to improve MOTL, consisting of tests about the course contents prepared previously by professors using the Moodle platform so students could do them during the class individually or in groups, with the assistance of the professors and thus receiving immediate results. H02, individual practical exercise, combined with H05, project basic learning (PBL), were the most used alternatives in HOTL, as a type of PBL specifically designed for large groups. They consisted of a brief PBL the students had to solve individually based on the class contents. For instance, they had to solve a construction solution for a specific architectural work in due course by detailing in hand on a small piece of paper like they were an architect on site solving their building team queries in real time.